# ADDITION IS ALL YOU NEED
# FOR ENERGY-EFFICIENT LANGUAGE MODELS

## ABSTRACT

Large neural networks spend most computation on floating point tensor multiplications. In this work, we find that a floating point multiplier can be approximated by one integer adder with high precision. We propose the linear-complexity multiplication ($\mathcal{L}$-Mul) algorithm that approximates floating point number multiplication with integer addition operations. The new algorithm costs significantly less computation resource than 8-bit floating point multiplication but achieves higher precision. Compared to 8-bit floating point multiplications, the proposed method achieves higher precision but consumes significantly less bit-level computation. Since multiplying floating point numbers requires substantially higher energy compared to integer addition operations, applying the $\mathcal{L}$-Mul operation in tensor processing hardware can potentially reduce **95%** energy cost by element-wise floating point tensor multiplications and **80%** energy cost of dot products. We calculated the theoretical error expectation of $\mathcal{L}$-Mul, and evaluated the algorithm on a wide range of textual, visual, and symbolic tasks, including natural language understanding, structural reasoning, mathematics, and commonsense question answering. Our numerical analysis experiments agree with the theoretical error estimation, which indicates that $\mathcal{L}$-Mul with 4-bit mantissa achieves comparable precision as `float8_e4m3` multiplications, and $\mathcal{L}$-Mul with 3-bit mantissa outperforms `float8_e5m2`. Evaluation results on popular benchmarks show that directly applying $\mathcal{L}$-Mul to the attention mechanism is almost lossless. We further show that replacing all floating point multiplications with 3-bit mantissa $\mathcal{L}$-Mul in a transformer model achieves equivalent precision as using `float8_e4m3` as accumulation precision in both fine-tuning and inference.

## 1 INTRODUCTION

Modern artificial intelligence (AI) systems are significant energy consumers. Because of the large scale computation needed for neural network inference, AI applications based on such models are consuming a considerable amount of electricity resource. Reportly, the average electricity consumption of ChatGPT service in early 2023 was 564 MWh per day, equivalent to the total daily electricity usage of 18,000 families in the United States[1]. It is estimated that Google's AI service could consume as much electricity as Ireland (29.3 TWh per year) in the worst-case scenario (de Vries, 2023).

Reducing the amount of computation needed by neural networks is the key to reduce both energy consumption and inference speed for large-scale AI models. Neural networks, especially large language models (LLMs) (Radford et al., 2019; Brown, 2020; Achiam et al., 2023; Touvron et al., 2023; Team et al., 2023), contain a large number of floating point parameters involved in element-wise and matrix multiplication computations. In transformer (Vaswani, 2017) based LLMs, the attention mechanism is a major bottleneck that limits the computation efficiency. Given a input context of $N$ tokens, the complexity of standard attention mechanism computation is $O(N^2)$, involving multiplying high dimensional tensors. Besides attention, there are also a large amount of element-wise multiplication and linear transformation computations. In this work, we propose the linear-complexity multiplication ($\mathcal{L}$-Mul) algorithm, which approximates floating point multiplication with integer addition operations. The algorithm can be integrated into existing models at various

---

[1]`https://www.eia.gov/tools/faqs/faq.php?id=97`

levels, such as replacing the multiplication in the attention mechanism or substituting all matrix and element-wise multiplications.

The proposed $\mathcal{L}$-Mul method will lead to a significantly reduced energy consumption for both model training and inference. In modern computing hardware, multiplications between floating point numbers consumes significantly higher energy than addition operations (Horowitz, 2014). Specifically, multiplying two 32-bit floating point numbers (`fp32`) costs four times energy as adding two `fp32` numbers, and 37 times higher cost than adding two 32-bit integers (`int32`). The rough energy costs for various operations are shown in Table 1. In PyTorch (Paszke et al., 2019), the default precision for accumulating tensor multiplication results is set to `fp32`. While I/O and control operations are not considered, approximating `fp32` multiplications with `int32` additions consumes only $1/37 \approx 2.7\%$ of the energy. When the accumulation precision is reduced to `fp16`, integer addition consumes approximately $4.7\%$ of the energy required for floating-point multiplication.

| Operation | Integer | | Floating Point | |
|---|---|---|---|---|
| | 8-bit | 32-bit | 16-bit | 32-bit |
| Addition | 0.03 pJ | **0.1 pJ** | 0.4 pJ | 0.9 pJ |
| Multiplication | 0.2 pJ | 3.1 pJ | 1.1 pJ | **3.7 pJ** |

Table 1: Energy cost of various arithmetic operations cited from Horowitz (2014).

We evaluate the numerical precision of $\mathcal{L}$-Mul algorithm on transformer-based language models with a wide range of language and vision tasks. Experiments with full-precision model weights show that replacing standard multiplication operations with $\mathcal{L}$-Mul in the attention mechanism is almost lossless for transformer-based LLMs. On natural language reasoning tasks, the average performance loss of $\mathcal{L}$-Mul-based attention is $0.07\%$ across commonsense, structured reasoning, language understanding. On vision tasks, $\mathcal{L}$-Mul-based attention gained $0.12\%$ accuracy improvement on visual question answering, object hallucination, and free-form visual instruction tasks. The experiment results are obtained by directly adapting pretrained LLMs with the standard attention implementation to the new $\mathcal{L}$-Mul-based attention mechanism without any additional training.

The error estimation and ablation study show that under the training-free setting, $\mathcal{L}$-Mul with 4-bit mantissa can achieve comparable precision as multiplying `float8_e4m3` numbers, and $\mathcal{L}$-Mul with 3-bit mantissa outperforms `float8_e5m2` multiplication. We also show that fine-tuning can fix the performance gap between $\mathcal{L}$-Mul and the standard multiplication. Fine-tuning a model where all multiplication operations in attention mechanisms, linear transformations, and element-wise products are replaced by 3-bit-mantissa $\mathcal{L}$-Mul results in comparable performance to fine-tuning a standard model with an accumulation precision of `float8_e4m3`.

In the expansive landscape of AI efficiency research, our approach centers on enhancing the efficiency of tensor arithmetic algorithms—a direction that is orthogonal yet complementary to prevailing efforts in I/O and control optimization (Jouppi et al., 2017; Choquette et al., 2021; Abts et al., 2022). We believe that truly energy- and compute-efficient AI computation will emerge from a holistic integration of optimizations across I/O, control, and arithmetic operations.

## 2 METHOD

### 2.1 BACKGROUND: FLOATING-POINT NUMBERS AND TENSORS

Most machine learning models, including neural networks, use floating point (FP) tensors to represent their inputs, outputs, and trainable parameters. Typical choices are 32-bit and 16-bit FP tensors (`fp32` and `fp16`) defined by the IEEE 754 standard shown in Figure 1.

Multiplication operations are generally more complicated than additions, and FP operation are more costly than integers (Horowitz, 2014). Table 1 shows that multiplying two `fp32` numbers consumes 37 times higher energy than adding two 32-bit integers. While the complexity of integer addition is $O(n)$ where $n$ is the number of bits used for representing the number, FP multiplication requires

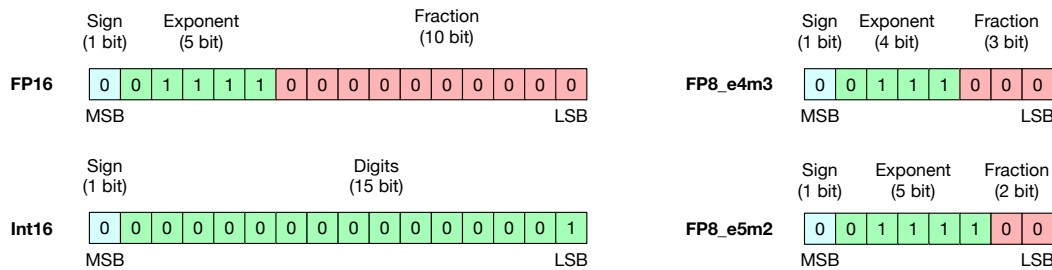

Figure 1: 16-bit, 8-bit floating point numbers defined in IEEE 754 and on various hardware for tensor computations, and the 16-bit integer. MSB stands for most significant bit and LSB stands for least significant bit.

$O(e)$ exponent addition, $O(m^2)$ mantissa multiplication, and rounding. Here $e$ and $m$ stand for the number of bits used for exponent and mantissa parts of the FP numbers.

Modern LLM training and inference involves a large number of FP calculations in tensor computation. Consider calculating the element-size and dot products of two 2-D tensors:

$$Y_1 = A \circ X, \ Y_2 = A \cdot X^T; \ A, X \in R^{(N,k)}$$

Calculating $Y_1$ involves $N^2$ FP multiplications (Mul). If $A$ and $X$ are both `fp32` tensors, $A \circ X$ consumes 37 times higher energy than adding two `int32` matrices of the save size. Similarly, Calculating $Y_2$ involves $(m \times n \times k)$ FP Mul and the same number of FP additions (Add). When $A$ and $X$ are `fp32` tensors, each Mul-Add operation for two numbers consumes $0.9 + 3.7 = 4.6$ (pJ) energy. If we replace the fp32 Mul with `int32` Add, the energy cost becomes $0.1 + 0.9 = 1.0$ (pJ), only 21.7% of the original cost. Similarly, if the inference is conducted in `fp16`, replacing `fp16` Mul with `int16` Add result in a $1 - (0.05 + 0.4)/(1.1 + 0.4) = 70\%$ energy saving.

## 2.2 LINEAR-COMPLEXITY MULTIPLICATION ($\mathcal{L}$-MUL)

We propose $\mathcal{L}$-Mul, a FP multiplication algorithm with $O(n)$ complexity, where $n$ is the bit size of its FP operands. Consider two FP numbers $x, y$, whose exponents and fractions are $x_e, y_e$ and $x_m, y_m$ respectively, the vanilla FP Mul result is

$$Mul(x,y) = (1 + x_m) \cdot 2^{x_e} \cdot (1 + y_m) \cdot 2^{y_e} = (1 + x_m + y_m + x_m \cdot y_m) \cdot 2^{x_e + y_e}$$

plus an `xor` operation ($\oplus$) to decide the sign of the result. Assume $x_m$ and $y_m$ are mantissas of $m$ bits. The $O(m^2)$ mantissa multiplication operation is the complexity bottleneck of this calculation. We remove this operation and introduce a new multiplication algorithm that processes mantissas with a computational complexity of $O(m)$:

$$\mathcal{L}\text{-Mul}(x,y) = (1 + x_m + y_m + 2^{-l(m)}) \cdot 2^{x_e + y_e}, \ \ l(m) = \begin{cases} m & \text{if } m \le 3, \\ 3 & \text{if } m = 4, \\ 4 & \text{if } m > 4. \end{cases} \tag{1}$$

The offset exponent $l(m)$ is defined according the observation shown in Figure 3. In the following sections, we show that (1) the $\mathcal{L}$-Mul operation can be implemented by integer Adders, and (2) the algorithm is more accurate and efficient than `fp8` multiplications.

The implementation of the algorithm is shown in Figure 2, where we also added the Inline PTX Assembly code we used to simulate the process on Nvidia GPUs. While Equation (1) contains 4 addition operations, the bit format design of FP numbers helps us implement the $\mathcal{L}$-Mul algorithm with one adder. Since the FP format handles $1 + x_m$ implicitly, we do not have to compute the value of $(1 + \dots)$. The integer addition operation also automatically send the mantissa carry to the exponent. If the mantissa sum is greater than 2, a carry is automatically added to the exponent. This is different from the rounding process in traditional FP multiplier, where the fraction is manually rounded to $1.x$ and the carry is added to the exponent as an independent step. As a result, the $\mathcal{L}$-Mul algorithm is more efficient than traditional FP multiplication by skipping both mantissa multiplication and rounding operations.

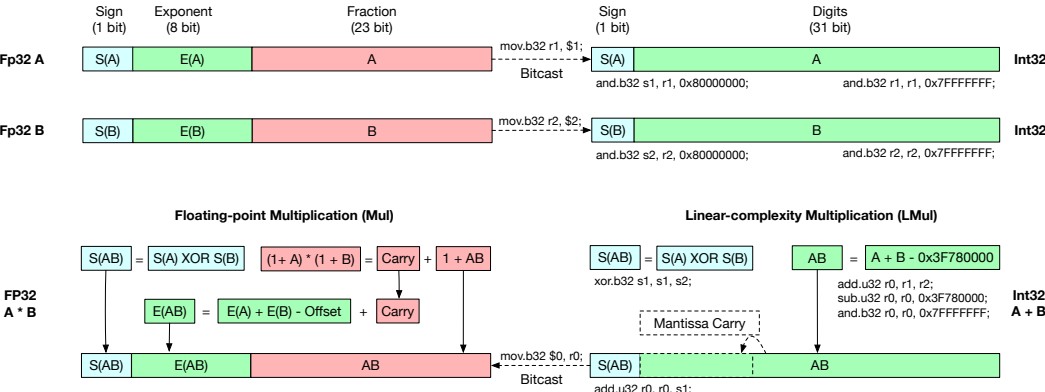

Figure 2: Coparing the process of regular floating-point multiplication and linear-complexity multiplication ($\mathcal{L}$-Mul) between two `fp32` numbers. In the inline PTX Assembly code, `$1` and `$2` are `fp32` registers storing inputs while `$0` is the `fp32` register for output. `s1, s2, r0, r1, r2` are unsigned int32 registers storing intermediate results. Note that the assembly program is only for numerical simulation on Nvidia GPUs. The optimal implementation is at the hardware level.

The construction of $\mathcal{L}$-Mul results can be expressed using the following equation, where all bit-level calculations are performed as operations between unsigned integers.

$$
\begin{aligned}
\mathcal{L}\text{-mul}(x, y)[0] &= x[0] \oplus y[0] \\
\mathcal{L}\text{-mul}(x, y)[1:] &= x[1:] + y[1:] - \text{offset}
\end{aligned}
\tag{2}
$$

We further implement the attention mechanism with $\mathcal{L}$-Mul. In transformer models, the attention mechanism has a high computation cost because of its $O(|C|^2)$ complexity to process the input context $C$. We found that $\mathcal{L}$-Mul can replace the complicated tensor multiplications with minimal performance loss needing no additional training. In this work we implement a more efficient attention mechanism as follows,

$$
K = H \cdot W_k, \; Q = H \cdot W_q, \; V = H \cdot W_V
$$

$$
A = softmax\left[\frac{\mathcal{L}\text{-matmul}(Q, K^T)}{\sqrt{d}}\right], \; H' = \mathcal{L}\text{-matmul}(A, H)
\tag{3}
$$

where $\mathcal{L}$-matmul$(Q, K^T)$ stands for a matrix multiplication operation where all regular FP multiplications are implemented in $\mathcal{L}$-Mul. By doing this, all FP multiplications are replaced with integer additions, which consumes significantly lower computation resource.

## 2.3 PRECISION AND COST ANALYSIS

In this section, we show that $\mathcal{L}$-Mul is more precise than `fp8_e4m3` multiplications but uses less computation resource than `fp_e5m2`. To be concise, we do not consider the rounding to nearest even mode in both error analysis and complexity estimation for both Mul and $\mathcal{L}$-Mul.

### 2.3.1 PRECISION ESTIMATION

The goal of the precision analysis is to find the precision of the $\mathcal{L}$-Mul algorithm is equivalent to rounding the fraction of a FP number to how many bits, e.g., `fp8` with 2- or 3-bit mantissas (`e5m2` or `e4m3`). Consider positive FP numbers $x = (1 + x_m) \cdot 2^{x_e}$ and $y = (1 + y_m) \cdot 2^{y_e}$, they can be written in the following format if we explicitly highlight the $k$ bits to be kept after rounding:

$$
x = (1 + x_k + x_r) \cdot 2^{x_e}, \;\; x' = (1 + x_k) \cdot 2^{x_e}
$$

$$
y = (1 + y_k + y_r) \cdot 2^{y_e}, \;\; y' = (1 + y_k) \cdot 2^{y_e}
$$

where $x_k, y_k$ are the first k bits of $x_m, y_m$, and $x_r, y_r$ are the value of remaining bits that will be ignored after the k-bit rounding. $x', y'$ are the rounded value of $x, y$ by keeping the first k bits of

the mantissa. Consider $x$ and $y$ has $m$-bit mantissa in their full precision. For example, `Float16` numbers have 10-bit mantissa and `BFloat16` contain 7 bits. The error of $Mul(x,y) = x \cdot y$ and its expectation are

$$e_{mul}^k = Mul(x,y) - Mul(x',y') = (x_k y_r + y_k x_r + x_r + y_r + x_r y_r) \cdot 2^{x_e + y_e}$$
$$E[e_{mul}^k] = f_1(m,k) \cdot E[2^{x_e + y_e}] \tag{4}$$

Comparing with a $k$-bit mantissa FP multiplication, the error of $k$-bit mantissa $\mathcal{L}$-Mul is

$$e_{lmul}^k = e_{mul}^k + (x_k y_k - 2^{-l(k)}) \cdot 2^{x_e + y_e}$$
$$E[e_{lmul}^k] = E[e_{mul}^k] + E[x_k\, y_k - 2^{-l(k)}] \cdot E[2^{x_e + y_e}] \tag{5}$$

With the equations above, we can compute the expectation of the precision gap between $k$-bit $\mathcal{L}$-Mul and FP multiplication:

$$E[e_{lmul}^k] - E[e_{mul}^k] = f_2(k) \cdot E[2^{e_x + e_y}], \quad E[e_{lmul}^k] = [f_1(m,k) + f_2(k)] \cdot E[2^{e_x + e_y}]$$

When $x_m, y_m$ are evenly distributed, we can calculate the following expectations,

$$E[x_k] = \frac{1}{2}(1 - 2^{-k}), \ E[x_r] = \frac{1}{2}(2^{-k} - 2^{-m})$$

By estimating $f_1(m,k)$ and $f_2(k)$ and further inferring $E[e_{lmul}^k]$ and $E[e_{mul}^k]$, we find that $\mathcal{L}$-Mul is more accurate than `fp8_e5m2` with evenly distributed operands. However, the weight distribution is often biased in pretrained LLMs. Based on the combined weight distribution of five popular LLMs, we find that $\mathcal{L}$-Mul can achieve higher precision beyond `fp8_e4m3` with 5-bit mantissa operands in practice. We support both claims with estimated errors detailed in Appendix A.

### 2.3.2 GATE COMPLEXITY ESTIMATION

In this section, we make a rough estimation for the amount of gate-level computations needed by $\mathcal{L}$-Mul and `fp8` multiplications. Multiplying two `fpn_eimj` number require the following computation: sign prediction, exponent addition with offset, a $j + 1$-bit mantissa multiplication, and exponent rounding. The mantissa multiplication includes $(j + 1)^2$ AND operations, 3 half adders and $2j - 2$ full adders. The exponent rounding needs $i$ half adders. In a regular circuit design, a full adder involves 2 AND, 2 XOR, and 1 OR. Each XOR has 4 NAND gates. As a result, a full adder consumes 11 gate-level computation, while a half adder (no incoming carry) consumes 5 gate-level computations (1 AND and 1 XOR).

In conclusion, the total amount of gate-level computation needed by `fp8` Mul can be estimated as

$$N_{\text{fp16}}^{\times} \approx 584, \ N_{\text{fp8-e4m3}}^{\times} \approx 325, \ N_{\text{fp8-e5m2}}^{\times} \approx 296 \tag{6}$$

$\mathcal{L}$-Mul consumes 1 XOR for sign prediction, 1 half adder, and $k - 2$ full adders. The total gate count needed by 16-bit and 8-bit $\mathcal{L}$-Mul can be estimated as follows,

$$N_{eimj}^{\mathcal{L}\text{-mul}} = N_1^{\oplus} + N_{int(i+j)}^{+} + N_{int8}^{+}$$
$$N_{\text{fp16}}^{\mathcal{L}\text{-mul}} \approx 256, \ N_{\text{fp8}}^{\mathcal{L}\text{-mul}} \approx 157 \tag{7}$$

$\mathcal{L}$-Mul with `fp8_e4m3` and `fp8_e5m2` operands have similar complexity since exponent offsets are typically implemented by 8-bit unsigned integer adders. As estimated, `fp16` $\mathcal{L}$-Mul requires less gates than `fp8` multiplications, and `fp8` $\mathcal{L}$-Mul is significantly more efficient.

To summarize the error and complexity analysis, $\mathcal{L}$-Mul is both more efficient and more accurate than `fp8` multiplication.

## 3 EXPERIMENTS

To prove the theoretical precision estimation and find out how $\mathcal{L}$-Mul-based LLMs perform on real tasks, we conducted experiments on various benchmarks with different transformer-based large language models. We evaluated `Llama-3.1-8b-Instruct` (Dubey et al., 2024),

`mistral-7b-v0.3-Instruct` (Jiang et al., 2023), `Gemma2-2b-It` (Team et al., 2024), and `Llava-v1.5-7b` (Liu et al., 2024) models, and found that the proposed method can replace different modules in transformer layers under fine-tuning or training-free settings. In this section, we first introduce the benchmarks and tasks used for evaluation, then compare the numerical error of the $\mathcal{L}$-Mul algorithm against models with `fp8` parameters. We also report the benchmarking results of LLMs under different precision settings.

### 3.1 TASKS

**Massive Multitask Language Understanding (MMLU)** (Hendrycks et al., 2020) contains 57 multi-choice natural language understanding tasks covering various high-school and college subjects. With 5 few-shot examples, the LLMs for evaluation are required to find the most appropriate answer option to each question. The benchmark focuses on evaluating the language understanding and knowledge abilities related to given subjects.

**BigBench-Hard (BBH)** (Srivastava et al., 2023) contains a set of complex symbolic tasks to evaluate the structural and logic reasoning abilities of language models. In this work, we select a subset of 17 multi-choice tasks to evaluate Llama and Mistral LLMs. We evaluate language models under the few-shot prompting setting for all BBH tasks.

**Common Sense.** We put together a set of 5 question answering tasks to evaluate the commonsense knowledge reasoning ability of LLMs. The set of task includes ARC-Challenge (Clark et al., 2018), CSQA (Saha et al., 2018), OBQA (Mihaylov et al., 2018), PIQA (Bisk et al., 2020), and SIQA (Sap et al., 2019), covering different aspects of factual and social knowledge.

**Visual Question Answering.** We select a set of multi-choice question answering tasks based on images for evaluating both vision and language understanding abilities of visual language models. The tasks include VQAv2 (Goyal et al., 2017), VizWiz (Gurari et al., 2018), and TextVQA (Singh et al., 2019), containing both unanswerable and answerable questions with different types of answers.

**Visual Instruction following.** We test the instruction following ability of `Llava-1.5-7b` model with the Llava-bench task (Liu et al., 2024) by generating free-form responses given images and corresponding instructions. Following the official evaluation guide, we evaluate the instruction following quality with GPT4o and compare the relative performance.

**Object Hallucination.** We explore if conducting inference with lower precision infects the truthfulness of the Llava model using the POPE benchmark (Li et al., 2023), which prompt visual language models with a sequence of yes/no questions about positive and negative objects.

**GSM8k** (Cobbe et al., 2021) consists of 8.5k human-crafted grade school math problems, with a test split of 1,000 problems designed to evaluate the arithmetic capabilities of language models. We conduct experiments on GSM8k in two different settings. In the training-free setting, we assess LLMs with few-shot, chain-of-thought prompting (Wei et al., 2022). Additionally, we fine-tune the Gemma2-2b-It model on the training split and evaluate its performance in a zero-shot setting.

### 3.2 PRECISION ANALYSIS

**Selection of $l(k)$.** We first visualize the mean square errors obtained by different $l(k)$ selections with different models on the GSM8k dataset in Figure 3. In the plot, we highlight the $l(k)$ configurations that leads to lower average error than `float8_e4m3` multiplications in model inference in red, and the $k, l(k)$ combinations leading to an error between `e4m3` and `e5m2` are underlined. In both models, $\mathcal{L}$-Mul with 3-bit mantissas is more accurate than `fp8_e5m2` and $\mathcal{L}$-Mul with 4-bit mantissas achieves comparable or lower error than `fp8_e4m3`.

**Mantissa size.** In section 2.3.1, we argued that the error expectation of $\mathcal{L}$-Mul can be lower than multiplying `fp8_e4m3` multiplication while using less computation resource than multiplying `fp8_e5m2` numbers. We hereby confirm the correctness of our theoretical precision estimates for the $\mathcal{L}$-Mul algorithm with experimental analysis. The average errors of Llama and Gemma models are illustrated in Figure 4.

The experiments demonstrated that across various sizes of LLMs, the $\mathcal{L}$-Mul algorithm using 6-bit mantissa FP operands approximates the lowest average error, significantly outperforming both `fp8`

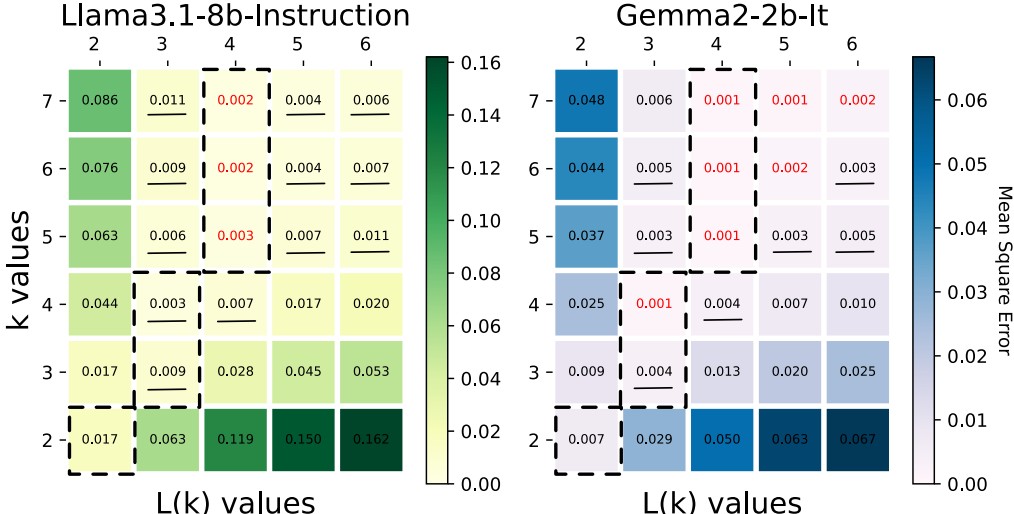

Figure 3: Mean square errors obtained by different $l(k)$ selections on Llama and Gemma models. The combinations achieving higher precision than `fp8_e4m3` are highlighted in read, and those outperforming `fp8_e5m2` are underlined. When $k = 4$ and $l(k) = 3$, the average error of the llama model is slighly lower but very close to `fp8_e4m3`.

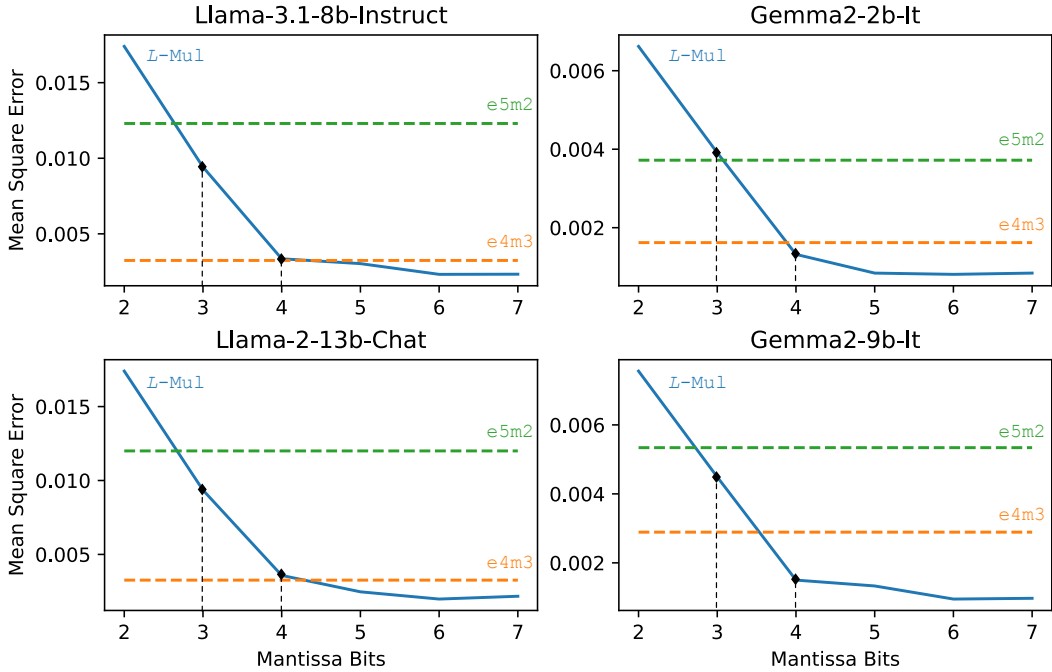

Figure 4: Comparing the error levels of linear-complexity multiplication ($\mathcal{L}$-Mul) against the number of mantissa bits comparing with 8-bit FP multiplication operation in different formats.

formats. Additionally, the 3- and 4-bit mantissa $\mathcal{L}$-Mul achieved accuracy on par with or exceeding that of `fp8_e5m2` and `fp8_e4m3` multiplication operations, respectively.

In the IEEE 754 format (with a 1-bit sign and a 5-bit exponent), using a 6-bit mantissa is equivalent to rounding `fp16` numbers down to `fp12`. By applying the complexity estimation method outlined in Equation (7), we can compute the gate count for 12-bit $\mathcal{L}$-Mul operations as follows:

$$N_{12}^{\mathcal{L}\text{-mul}} \approx 201 < N_{fp8}^{\times} \approx 300 \tag{8}$$

The experimental results further confirm that $\mathcal{L}$-Mul is more efficient and accurate than `fp8` multiplication. Although we estimated gate counts as an indicator of computational complexity, the actual difference in energy cost is greater than the complexity gap suggests. Based on the energy consumption reported in Horowitz (2014), an `fp8` multiplication consumes approximately 0.25 pJ to 0.4 pJ, while a 16-bit $\mathcal{L}$-Mul uses around 0.06 pJ of energy.

## 3.3 BENCHMARKING

In this section, we demonstrate that $\mathcal{L}$-Mul can replace tensor multiplications in the attention mechanism without any loss of performance, whereas using `fp8` multiplications for the same purpose degrades inference accuracy. This indicates that we can achieve the same model inference performance while reducing the energy cost of attention computations by 80%. Additionally, we present the full-model fine-tuning performance when all tensor multiplication operations are replaced with $\mathcal{L}$-Mul on the GSM8k benchmark.

**Textual tasks.** Table 2 presents the evaluation results of the Llama and Mistral models on various natural language benchmarks, including MMLU, BBH, ARC-C, CSQA, PIQA, OBQA, and SIQA. In these experiments, the matrix multiplications in the attention layers, both before and after the softmax operation, were replaced with 8-bit tensor computations in different formats or $\mathcal{L}$-Matmul following the implementation we discussed in Equation (3).

| Precision | BBH | MMLU | ARC-R | CSQA | OBQA | PIQA | SIQA | Avg. |
|---|---|---|---|---|---|---|---|---|
| | | | Mistral-7b-Instruction-v0.3 | | | | | |
| BFloat16 | 55.85 | 62.20 | 75.94 | 71.42 | 76.20 | 80.74 | 44.83 | 69.83 |
| Float8_e4m3 | 55.16 | 62.18 | 75.39 | 71.25 | 76.00 | 80.47 | 44.63 | 69.55 |
| Float8_e5m2 | 53.20 | 61.75 | 74.91 | 71.25 | 74.40 | 79.76 | 44.52 | 68.97 |
| $\mathcal{L}$-Mul | 55.87 | 62.19 | 76.11 | 71.09 | 76.60 | 80.52 | 45.34 | 69.93 |
| | | | Llama-3.1-8B-Instruct | | | | | |
| BFloat16 | 70.79 | 68.86 | 82.51 | 74.53 | 84.20 | 84.00 | 45.96 | 74.24 |
| Float8_e4m3 | 69.91 | 68.16 | 81.66 | 74.28 | 82.20 | 83.51 | 45.34 | 73.40 |
| Float8_e5m2 | 62.94 | 66.61 | 80.12 | 73.30 | 79.40 | 81.07 | 45.39 | 71.86 |
| $\mathcal{L}$-Mul | 70.78 | 68.54 | 82.17 | 74.28 | 84.20 | 83.30 | 46.06 | 74.00 |

Table 2: Comparing the attention mechanism implemented with 16- and 8-bit tensor multiplication operations and $\mathcal{L}$-Mul approximations. Note that the $\mathcal{L}$-Mul computations cost significantly less resource than `fp8` tensors.

The results indicate that $\mathcal{L}$-Mul not only requires significantly fewer computational resources but also delivers higher precision than `float8-e4m3` tensors in 12 out of 14 experiments using Mistral and Llama models. This leads to a minimal performance gap when compared to `bf16` inference. On average, across the two models, the performance difference between `bf16` and $\mathcal{L}$-Mul is just 0.07%. These findings suggest that matrix multiplication operations in the attention mechanism can be seamlessly replaced with the $\mathcal{L}$-Mul algorithm without any loss of accuracy or the need for additional training.

**GSM8k.** We evaluated the performance of three language models—Mistral-7b-Instruct-v0.3, Llama3.1-7b-Instruct, and Gemma2-2b-It—on the GSM8k dataset using few-shot prompting and $\mathcal{L}$-Mul-based attention. The models were tested under different numerical precision formats: `bf16`, `fp8_e4m3`, `fp8_e5m2`, and the $\mathcal{L}$-Mul method. The results are summarized in Table 3.

Notably, the $\mathcal{L}$-Mul-based attention mechanism slightly improved the average performance compared to the `bf16` baseline. Mistral-7b-Instruct-v0.3 and Gemma2-2b-It both exhibited improved accuracies with $\mathcal{L}$-Mul, achieving 52.92% and 47.01% respectively. Llama3.1-7b-Instruct's accuracy with $\mathcal{L}$-Mul was slightly lower than its `bf16` performance but still higher than with `fp8_e4m3` and `fp8_e5m2`. On contrary, rounding the tensors in the attention computation to `fp8_e5m2` leads to a significant performance drop although it's more complicated than $\mathcal{L}$-Mul.

| Model | Bfloat16 | Float8_e4m3 | Float8_e5m2 | $\mathcal{L}$-Mul |
|---|---|---|---|---|
| Mistral-7b-Instruct-v0.3 | 52.54 | 52.39 | 50.19 | 52.92 |
| Llama3.1-7b-Instruct | 76.12 | 75.44 | 71.80 | 75.63 |
| Gemma2-2b-It | 45.87 | 45.94 | 44.43 | 47.01 |
| **Average** | 58.17 | 57.92 | 55.47 | 58.52 |

Table 3: GSM8k accuracy with Mistral, Llama, and Gemma models with few-shot prompting and attention mechanism implemented in different precision levels.

**Vision-language tasks.** The performance of the `Llava-v1.5-7b` model on VQA, object hallucination, and instruction following tasks are shown in Table 4. Similar to the experiments on language tasks, the attention computation is conducted with different precision/methods while other linear transformation layers are unchanged. Except for TextVQA where the accuracy gap is $0.5\%$, the performance of $\mathcal{L}$-Mul and BFloat16 attentions are comparable. The VQA tasks are evaluated with the official evaluation scripts and the Llava-Bench results are generated by GPT-4o.

| Task | POPE | | | | Llava-Bench | | | | TextVQA |
|---|---|---|---|---|---|---|---|---|---|
| **Split** | rand. | adv. | pop. | all | comp. | conv. | detail. | all | all |
| BFloat16 | 86.20 | 83.17 | 85.13 | 84.83 | 66.80 | 57.60 | 41.40 | 57.50 | 57.90 |
| $\mathcal{L}$-Mul | 86.57 | 83.19 | 85.34 | 85.03 | 64.90 | 58.70 | 43.30 | 57.50 | 57.41 |
| **Task** | VQAv2 | | | | VizWiz | | | | |
| **Split** | yes/no | num. | other | all | yes/no | num. | unans. | other | all |
| BFloat16 | 91.88 | 59.04 | 70.56 | 78.03 | 77.19 | 45.24 | 71.75 | 38.19 | 49.31 |
| $\mathcal{L}$-Mul | 91.78 | 58.93 | 70.73 | 78.06 | 78.54 | 50.48 | 73.78 | 38.41 | 50.16 |

Table 4: Evaluating the performance of different attention implementation on the `Llava-v1.5-7b` model. VQAv2, VizWiz, and TextVQA are visual question answering tasks, POPE evaluates object hallucination, and Llava-Bench assesses the instruction following ability scored by GPT-4o.

**$\mathcal{L}$-Mul with fewer bits.** In this section, we explore how $\mathcal{L}$-Mul-based attention precision influences the overall model performance using the MMLU benchmark with Mistral and Llama models. We implement the attention mechanism with $\mathcal{L}$-Mul and only keep the first $k$ bits of the operand tensors. The results of $\mathcal{L}$-Mul attention with different precision are listed in Table 5. As expected, using $\mathcal{L}$-Mul with a 4-bit mantissa achieves performance comparable to or slightly better than that of `bf16` and `fp8_e4m3`. However, performance drops proportionally to the estimated error depicted in Figure 4. When $k = 3$, both models significantly outperform their `fp8_e5m2` counterparts, with the Llama model's performance remaining close to that of `fp8_e4m3`. When $k = 2$, the Llama model's performance is comparable to that of `fp8_e5m2` rounding. This suggests that with the Llama model, we can perform $\mathcal{L}$-Mul directly on `fp8` models without compromising performance.

| Model | e4m3 | e5m2 | $k=4$ | $k=3$ | $k=2$ |
|---|---|---|---|---|---|
| Mitral | 62.18 | 61.75 | 62.16 | 62.06 | 61.08 |
| Llama | 68.16 | 66.61 | 68.43 | 68.12 | 66.67 |

Table 5: The performance of Mistral models with attention mechanism implemented with $k$-bit tensor $\mathcal{L}$-Mul.

| 8bit Acc. | e4m3 | e5m2 | $\mathcal{L}$-Mul |
|---|---|---|---|
| GSM8k | 36.09 | 7.96 | 37.91 |

Table 6: Zero-shot fine-tuned Gemma2-2b models with 8-bit accumulation precision. $\mathcal{L}$-Mul uses `fp8_e4m3` inputs.

**Full-model fine-tuning.** To further explore the impact of the $\mathcal{L}$-Mul algorithm, we go beyond implementing attention layers with $\mathcal{L}$-Mul by replacing all multiplication operations—including matrix multiplications in linear transformations, element-wise multiplications, and those within attention layers—with `fp8_e4m3` $\mathcal{L}$-Mul for the `Gemma2-2b-It` model. We then fine-tune the updated model on the training set of the GSM8k corpus and evaluate both the fine-tuned `fp8` and $\mathcal{L}$-Mul

models under a zero-shot setting on the GSM8k test set. Note that the $\mathcal{L}$-Mul operations in this experiment takes operands with 3-bit mantissas ($k = 3$) and the accumulation precision is `fp8_e4m3` to explore an extremely efficient setting.

The experimental results demonstrate that a fine-tuned `fp8_e4m3` $\mathcal{L}$-Mul model achieves performance comparable to a standard fine-tuned `fp8_e4m3` model under `fp8` accumulation precision. This suggests that $\mathcal{L}$-Mul can enhance training efficiency without compromising the fine-tuned model's performance. Moreover, it reveals the potential of training $\mathcal{L}$-Mul native LLMs for accurate and energy-efficient model hosting.

## 4 RELATED WORK

Reducing the computation needed by neural networks while maintain the performance is an important problem which entailed multiple research directions. Typical methods include neural network pruning, quantization, and improved tensor I/O implementations.

**Pruning.** Neural network pruning focuses on improving the inference efficiency by reducing the number of connections among layers (Han et al., 2015a;b; Wang et al., 2020). Neural network pruning methods usually involves training. After important weights are identified, the neural networks are re-trained to further update the selected weights for specific tasks. Different from model pruning, the method we proposed is designed for general tasks, requiring no task-specific re-training.

**Optimizing tensor I/O.** On regular GPUs, moving tensors between GPU SRAM and high-bandwidth memory (HBM) is the main bottleneck of time and energy consumption. Reducing the I/O operations in transformer models and making the best use of the HBM can significantly improve the efficiency of AI training and inference (Dao et al., 2022; Dao; Kwon et al., 2023). Our method, which focuses on optimizing arithmetic operations, is orthogonal to this direction.

**Rounding and quantization.** Standard neural network weights are stored as 32-bit or 16-bit FP tensors. However, the full-sized weights takes a considerable amount of GPU memory. To improve the storage efficiency, both weights storage and computation can be conducted in a lower precision, for example, using 16-bit, 8-bit, or 4-bit FP and Int (fp16, bf16 (Kalamkar et al., 2019), fp8-e4m3, fp8-e5m2 (Micikevicius et al., 2023), int8 (Dettmers et al., 2022), fp4, and int4 (Dettmers et al., 2024)) tensors to represent model weights. Inference with lower-bit parameters usually hurts the computation accuracy and impacts the performance of pretrained models, and Integer-based quantization methods spend significant time to handle outlier weights. comparing to the quantization methods, our method requires less computation but achieves higher accuracy.

## 5 FUTURE WORK

To unlock the full potential of our proposed method, we will implement the $\mathcal{L}$-Mul and $\mathcal{L}$-Matmul kernel algorithms on hardware level and develop programming APIs for high-level model design. Furthermore, we will train textual, symbolic, and multi-modal generative AI models optimized for deployment on $\mathcal{L}$-Mul native hardware. This will deliver high-speed and energy-efficient AI hosting solutions, reducing the energy cost for data centers, robotics, and a wide spectrum of edge-computing devices.

## 6 CONCLUSION

In this paper, we introduced $\mathcal{L}$-Mul, an efficient algorithm that approximates floating-point multiplication using integer addition. We first demonstrated that the algorithm exhibits linear complexity relative to the bit size of its floating-point operands. We then showed that the expected accuracy of $\mathcal{L}$-Mul surpasses that of `fp8` multiplications while requiring significantly less computational power. To assess the practical impact of $\mathcal{L}$-Mul, we evaluated it on natural language, vision, and mathematics benchmarks using popular language models. Our experiments indicate that $\mathcal{L}$-Mul outperforms 8-bit transformers with lower computational consumption and achieves lossless performance when applied to computation-intensive attention layers without additional training. Based on this evidence, we argue that tensor multiplications in language models can be effectively implemented using $\mathcal{L}$-Mul to preserve performance while enabling energy-efficient model deployment.

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

## A  ERROR ESTIMATION

We calculate the error expectations with different $(n, k)$ combinations as follows in Table 7. The values are calculated with the actual parameters of Mistral, Llama, and Gemma models. For even distribution, we use the expectations introduced in Section 2.3.1. For real distribution, we estimate the average value of possible operands using the parameters of five popular pretrained LLMs.

| K values | | 1 | 2 | 3 | 4 | 5 | 6 |
|---|---|---|---|---|---|---|---|
| Even Distribution | $abs[f_1(n=7,k)]$ | 0.68 | 0.35 | 0.17 | 0.081 | 0.035 | 0.012 |
| | $abs[f_1(n=7,k)+f_2(k)]$ | 0.68 | 0.43 | 0.30 | 0.24 | 0.20 | 0.19 |
| Real Distribution | $abs[f_1(n=7,k)]$ | 0.61 | 0.33 | 0.16 | 0.077 | 0.033 | 0.011 |
| | $abs[f_1(n=7,k)+f_2(k)]$ | 0.16 | 0.18 | 0.18 | 0.12 | 0.15 | 0.14 |

Table 7: Average error expectation with five different language models on floating point multiplication and $\mathcal{L}$-Mul with different rounding representations when the full precision is `BFloat16`. $K$ stands for the bit number of the operand mantissa.

We find that when the operands are distributed evenly, $\mathcal{L}$-Mul is more accurate than `float8_e5m2` multiplications. However with real models, $\mathcal{L}$-Mul can achieve higher precision than `float8_e4m3` calculations.

