# OpenReview forum: "Addition is All You Need for Energy-efficient Language Models"
_ICLR.cc/2025/Conference — ICLR 2025 Conference Withdrawn Submission_

### Official Review · Reviewer_Yfgk · 2024-10-24

**Soundness:** 3
**Presentation:** 3
**Contribution:** 3
**Rating:** 5
**Confidence:** 3

**Summary:**

This paper proposes replacing the multi-operation process in large language models (LLMs) with summation operations, significantly reducing energy consumption.

**Strengths:**

This paper explores an effective approach to reducing the energy consumption of large language models (LLMs) by replacing complex multi-operations with more energy-efficient summation operations.

**Weaknesses:**

The primary concern is hardware support, as current GPUs are highly optimized for multiplication operations. If model inference relies solely on summation, a significant portion of the GPU's resources dedicated to multiplication would be underutilized, leading to inefficiency.

**Questions:**

It would be beneficial to present experimental results and comparisons on real hardware, showcasing the outcomes in terms of energy and power consumption. This would provide concrete evidence of the efficiency gains from the proposed approach.

---

### Official Review · Reviewer_XakH · 2024-10-29

**Soundness:** 2
**Presentation:** 3
**Contribution:** 2
**Rating:** 3
**Confidence:** 4

**Summary:**

Large neural networks dedicate the majority of their computation to floating-point tensor multiplications. This work approximates a floating-point multiplier using a high-precision integer adder.
Through theoretical derivation, it is shown that, compared to 8-bit floating-point multiplication, this method achieves higher precision while reducing computational demand. The application of the L-Mul operation can potentially reduce energy consumption by 95% for element-wise floating-point tensor multiplications and by 80% for dot product calculations.
Theoretical error expectations for L-Mul have been calculated, and the algorithm has been evaluated across textual, visual, and symbolic tasks. Results demonstrate that L-Mul with a 4-bit mantissa achieves comparable precision to float8 e4m3 multiplication, while L-Mul with a 3-bit mantissa outperforms float8 e5m2.

**Strengths:**

1. The method proposed in this paper, using integer adders in place of floating-point multipliers, is innovative and may support the reduction of energy consumption in large language models.
2. The experiments examine the method's effectiveness across a wide range of task types.
3. The paper is clearly presented, with visually appealing and easy-to-understand figures and tables that also exhibit a novel presentation style.

**Weaknesses:**

1. The paper contains several spelling errors, such as 'Reportly' which should be 'Reportedly,' 'of the save size' which should be 'same size,' 'highlighted in read' which should be 'in red,' as well as 'slighly' for 'slightly,' 'evenlly' for 'evenly,' and 'coparing' for 'comparing,' among others.
2. The selection of L(k) is based on Figure 3, which only includes experimental analysis for the Llama3.1-8b and Gemma2-2b models, making it difficult to establish the general applicability of the current L(k) values across different models.
3. The experimental section lacks model diversity, as it only includes Mistral-7b, Llama3.1-7b, and Gemma2-2b. Additional experiments could be conducted on models such as OPT and Mamba, as well as across different model scales, such as Llama3-13B and Llama3-70B.
4. There is a lack of hardware experiments, with metrics like gate complexity and power consumption only theoretically estimated. Power consumption calculations are based on results published in 2014, which provides limited guidance to the readers. It's suggested that the authors provide RTL implementations, synthesis, and simulation results to demonstrate the ideas.
5. From the perspective of hardware designers, similar approximation techniques have been widely discussed. The authors are required to provide more discussions and comparisons.
6. Some transitions lack clarity. For instance, moving from Equation (1) to Figure 2 requires readers to infer the offset transformation, which could be more explicitly explained.
7. The current experiments cover a variety of tasks, but the scope of applicability for the L-Mul algorithm is not clearly defined.

**Questions:**

1. The reviewer has some doubts regarding the gate count calculations. According to Section 2.3.2, the gate count for FP8-e5m2 should include four parts: sign prediction, exponent addition with offset, a (j + 1)-bit mantissa multiplication, and exponent rounding.
   For sign prediction, an XOR is required, using four gates; exponent addition with offset, even with all additions implemented as full adders, would require only 5 * 11 * 2 = 110 gates. Mantissa multiplication, as described in the first paragraph, requires (j + 1)^2 AND gates, three half adders, and 2j - 2 full adders, totaling 9 + 3 * 5 + 2 * 11 = 46 gates; exponent rounding requires five half adders, which requires 5 * 5 = 25 gates.
   Summing these, we get 4 + 110 + 46 + 25 = 185 gates, which is still far from the 296 gates stated in Equation 6.
   Can the method proposed in this paper indeed significantly reduce gate count? It's suggested that the authors provide a more accurate calculation method and the calculation details.
2. Additionally, the proposed L-Mul involves converting FP32 to int32, performing integer addition, and then converting the result back to FP32. The reviewer is curious about the communication and resource overhead and hopes for more experiments and profiling results. It seems the proposed method helps reduce the computation costs while doing no favor to the crucial communication bottlenecks of LLMs.

---

### Official Review · Reviewer_qgSz · 2024-11-02

**Soundness:** 2
**Presentation:** 3
**Contribution:** 2
**Rating:** 5
**Confidence:** 4

**Summary:**

This paper proposes the linear-complexity multiplication algorithm that approximates floating point multiplications with integer additions to save compute energy. Evaluations on various tasks show that the accuracy is comparable to FP8 multiplications.

**Strengths:**

Interesting idea that approximates floating-point multiplications with integer additions to reduce energy consumption.

**Weaknesses:**

While the proposed approximation method is interesting, the practical value of simply replacing floating-point multiplications with high-precision additions is not convincing, especially in the context of LLMs.

+ First, in an LLM serving system (software + hardware), it is known that the on-chip and off-chip data movement and storage (induced by weights and kv caches), rather than computations, dominate energy consumption. Although computations are streamlined, the proposed method introduces non-negligible overheads on memory footprint and data traffic (due to high-precision additions), which might offset the energy saving of computations.

+ Second, I think a system-level comparison with INT4 quantization on a real system (e.g., FPGA-based system) is helpful to demonstrate the efficiency of the proposed method, since the energy consumption of an INT4 MAC is also far less than that of FP, and the storage and bandwidth requirements of INT4 data can also be significantly reduced.

Moreover, the authors claim that the proposed method is orthogonal and complementary to methods that optimize I/O and control. However, evaluations of this (such as incorporation with weight/kv cache pruning) are missing.

**Questions:**

Please see the weaknesses.

---

### Official Review · Reviewer_CWXw · 2024-11-04

**Soundness:** 3
**Presentation:** 2
**Contribution:** 3
**Rating:** 5
**Confidence:** 4

**Summary:**

This paper presents L-Mul, an algorithm that approximates floating point multiplications with integer additions, performed at higher precision. The trade-off is favorable towards L-Mul against other data formats, as significant energy savings may be achieved thanks to the use of fewer gate-level computations, with each Add consuming less energy than Mul. The authors primary focus is replacing the attention matmuls with L-Mul. 12-bit L-Mul performance are compared against fp16, fp8_e4m3, fp8_e5m2. Results are reported on various LLM (llama-3.1-8b, mistral-7b-v0.3, gemma2-2b, llava-v1.5-7b) across several common NLP tasks.

**Strengths:**

- improving energy efficiency with limited or no accuracy degradation is a topic of great interest for both academia and industry
- to the best of my knowledge, the proposed approach is novel
- 12-bit L-Mul achieves better or comparable accuracies to FP8 formats across multiple tasks. Model and benchmark selection is relevant and appropriate, and provides convincing evidence on the accuracy claims insofar they relate to replacement of the attention matmuls
- energy estimates are more handwaved and reported in a confusing way, but I agree with the claim that there is an expectation of potentially significant energy savings, at least at the level the individual computations that are being replaced by L-Mul

**Weaknesses:**

- "is all you need" title format is generally overused and in this case does not even fairly represents the content of the paper: most experiments leave regular matmul in place everywhere across the network, with the exception of the attention matmuls. I strongly recommend the authors to use a more informative title for this paper
- a main limitation is that the matmul replacement (and related accuracy results) is primarily focused on attention modules. There is only one very limited example (Table 6) where L-Mul is applied across the full model. What reasons drove the authors to this apply L-Mul only to the attention, in the first place?
- discussion on energy savings is inconsistent and confusing:
	- 95% saving for element-wise computation mentioned in the abstract is not discussed further in the paper
	- 80% saving for dot product is mentioned at the beginning of Section 3.3 but not supported by evidence (it's unclear how this is derived, and for what precision of L-Mul)
	- it is not clear at all how these two numbers are derived, may the authors clarify this?
	- crucially, a discussion of energy saving *at model level* is not made. I would encourage the authors to provide such estimate
- energy savings are estimates (not measurements) and the behavior of the L-Mul algorithm is simulated, with the authors leaving as future work an actual implementation of this concept at hardware level
- the meaning of Fig 3 should be better explained: is this the combined mse of every matmul and every example of GSM8k, for the two given models? Or a subset of these combinations?
- the abstract mentions that L-Mul with 4 and 3 bits mantissa (= fp10 and fp9, respectively) achieve comparable performance to fp8_e4m3 and fp8_e5m2, respectively. This is reported in Fig. 3 and 4. However, most of the results (Table 2,3,4, possibly 6) relate to L-Mul with 6 bits mantissa (fp12). At least this is my interpretation of the text, the precision used across different experiments is not stated as clearly as it should have been, and I would encourage the authors to amend the text throughout to clarify this point
- the paper shows an embarrassing lack of proofreading: it is full of typos, minor errors, and does not use notation consistently. Some examples include:
	- a sentence is repeated in the *abstract* with slightly different wording
	- Section 2.1: element-size -> element-wise
	- Section 2.1: A and X matrices are defined to be (N,k) but these dimensions do not match the subsequent calculations:
		- Y1 element-wise product of (N,k) matrices involves N*k multiplications, not N^2
		- Y2 dot product uses N * N * k multiplications, not m * n * k (m was defined as mantissa bits); the number of additions is similar but not exactly the same
	- Section 2.1: a comparison is made between fp16 and int16 energy numbers but the origin of the int16 numbers is not reported (they are not provided in Table 1 along with the other formats)
	- Fig 2 caption: coparing --> comparing
	- Section 2.2 (and elsewhere): the distinction between L-Mul and L-MatMul is unclear and appears to be inconsistent. They are the same op applied to different part of the network. Reported results usually refer to L-Mul but in most cases the new op is applied to the attention matmuls (which eq. 3 calls L-MatMul). Future work talks about L-Mul and L-Matmul. Would recommend to use a single name, as the op is the same
	- Section 2.2: when l(m) is defined in Eq. 1, the text states that it "is defined according to observation in Figure 3" but at this point it is totally unclear what observations are being referred to
	- Section 2.3.1: the meaning of the first sentence is unclear, the grammar is incorrect
	- Section 2.3.2: the texts mentions "the total amount of gate-level computations needed by fp8 Mul" but reports fp16 as well
	- Eq. 4 uses k,r in the explicit form of e^k_mul but f1 is then expressed as function of m,k instead (m being k+r)
	- fp8_m4e3 and fp8_m5e2 baseline errors are used in Fig. 3 as reference but their value is only reported later in Fig. 4!
	- Fig 3 caption: read --> red
	- Section 3.2: "the experimental results further confirm that L-Mul is more efficient and accurate than FP8 multiplication" --> would probably remove this sentence. There aren't additional experiments on efficiency. Accuracy is reported in the following section
	- Appendix A uses n for number of mantissa bits that are called m everywhere else

**Questions:**

- why the focus on replacement of the attention matmul, instead of all matmul across the model?
- can the authors provide estimates of energy savings at model level?
- extensive proofreading is needed to improve readability and consistency

**Details Of Ethics Concerns:**

Not discussed in the paper. No concern on my end.

---

### Note · Authors · 2024-11-14

I have read and agree with the venue's withdrawal policy on behalf of myself and my co-authors.